# ^68^Ga-PSMA PET/CT in Recurrent Prostate Cancer after Radical Prostatectomy Using PSMA-RADS Version 2.0

**DOI:** 10.3390/diagnostics14121291

**Published:** 2024-06-19

**Authors:** Gabriele Masselli, Saadi Sollaku, Cristina De Angelis, Elisabetta Polettini, Gianfranco Gualdi, Emanuele Casciani

**Affiliations:** 1Department of Radiological Sciences, Oncology and Anatomo-Pathology, “Sapienza” University of Rome, Piazzale Aldo Moro, 5, 00185 Rome, Italy; gabriele.masselli@uniroma1.it; 2PET/CT Section, Pio XI Private Hospital, Via Aurelia 559, 00165 Rome, Italy; saadi.sollaku@gmail.com (S.S.); elisabettapolettini@alice.it (E.P.); gianfranco.gualdi@uniroma1.it (G.G.); emanuelecasciani@gmail.com (E.C.)

**Keywords:** prostate cancer, ^68^Ga-PSMA PET/CT, ^68^Ga-PSMA, biochemical recurrence, PSMA-RADS

## Abstract

Background: ^68^Ga-PSMA PET/CT is superior to standard-of-care imaging for detecting regional and distant metastatic recurrent prostate cancer. The objective of our study was to evaluate the performance of ^68^Ga-PSMAPET/CT in our patient population, using the new PSMA-RADS version 2.0. Methods: A total of 128 patients scanned with ^68^Ga-PSMA PET/CT for detection of recurrence after RP were analyzed with PSMA-RADS version 2.0. For the analysis of the detection rate, categories PSMA-RADS 3 to 5 were considered as “positive for malignancy” and 1–2 as “negative”. Results: According to PSMA-RADS v2.0, we classified patients as follows: 23 patients without PSMA-RADS because they were negative; PSMA-RADS 1: 10 patients; PSMA-RADS 2: 4 patients; PSMA-RADS 3A: 11 patients; PSMA-RADS 3B: 2 patients; PSMA-RADS 3C: 2 patients; PSMA-RADS 3D: 2 patients; PSMA-RADS 4: 13 patients; PSMA-RADS 5: 61 patients. Conclusions: The overall detection rate of ^68^Ga-PSMA PET/CT was 71%. By dividing the patients into fourgroups according to PSA level before examination, we obtained the following detection rates: PSA < 0.2 ng/mL 38%; 0.2 ≤ PSA < 0.5 ng/mL 57%; 0.5 ≤ PSA ≤ 1 ng/mL 77%; and PSA > 1 ng/mL 95%. Conclusion: Using PSMA-RADS version 2.0, we obtained detection rate values comparable with recent literature both in absolute terms and in relation to different PSA levels.

## 1. Introduction

Prostate cancer (PCa) is the second most common malignancy in the male population worldwide [1].Treatment options for localized PCa include radical prostatectomy (RP), external beam radiotherapy (EBRT), or brachytherapy. Unfortunately, 30–50% of patients develop biochemical recurrence (BR) in the first 10 years after RP [2]. BR is commonly defined as prostatic specific antigen (PSA) level above 0.2 ng/mL (0.2 μg/L) across two measures (and rising) after RP [3]. However, BR does not provide information on the disease site(s) or differentiate between a local, regional, or systemic recurrence, which is critical information for further appropriate management. Imaging can help in this setting. Conventional imaging modalities used to detect prostate cancer recurrence include computed tomography (CT), bone scan, magnetic resonance imaging (MRI), ^18^fluorine–cholinepositron emission tomography/computed tomography (^18^F-cho PET/CT) and more recently, ^68^gallium prostate-specific membrane antigen (^68^Ga-PSMA) PET/CT. ^68^Ga-PSMA PET/CT is superior to standard-of-care imaging for detecting regional and distant localizations in recurrent PCa [4]. ^68^Ga-PSMA PET/CT positivity increases with PSA level. The detection rate for a PSA level of 0.2–0.5 ng/mL was 28–57.9%, while, for a PSA level of 0.5–1.0 ng/mL, the detection rate was 39–72.7%. For PSA levels of 1.0–2.0 ng/mL, the detection rate was 64–93.1% and for PSA levels higher than 2.0 ng/mL the detection rate was 87.5–96.8% [5,6]. The purpose of our study was to assess the performance of ^68^Ga-PSMAPET/CT, using the new Prostate-Specific Membrane Antigen Reporting and Data System (PSMA-RADS) version 2.0 [7].

## 2. Materials and Methods

### 2.1. Patients

One hundred and seventy-one consecutive patients who underwent ^68^Ga-PSMA-11 PET/CT imaging for recurrent PCa were selected from our institution’s database (January 2019 to April 2022). Only patients who had undergone RP, who had not received chemotherapy, in whom PET findings were confirmed by imaging techniques or by clinical data were included. In total, 128 patients were included in this retrospective study. A total of 13 patients had formerly received androgen-deprivation therapy (ADT) (i.e., more than 7 months before PSMA-PET imaging), 40 patients had also received radiotherapy (RT) (prostatectomy bed only, *n* = 37; prostatectomy bed plus pelvic lymph nodes, *n* = 3), and 5 patients underwent pelvic lymphadenectomy. All patients gave their written informed consent for anonymized evaluation and publication of their data. In all patients, the serum PSA level at the time of the PET scan was available.

### 2.2. PET/CT Acquisition and Image Reconstruction

PET/CT scans were acquired on a Discovery IQ high-sensitivity PET/CT scanner (GE Healthcare, Technologies Inc., Chicago, IL, USA). Patients were hydrated, asked to void immediately before the acquisition, and scanned from vertex to mid-thigh approximately 60–70 min after intravenous administration. The administered activity of ^68^Ga-PSMA-11 was 2–4 MBq per kilogram body weight. PET images were obtained in the time-of-flight mode. A simultaneous CT scan was acquired using automatic mA-modulation, and 120 kV. The images were reconstructed using the iterative reconstruction technique. Attenuation correction was provided by CT data. Intravenous iodinated contrast medium was administered in selected cases, such as equivocal uptake in soft-tissue site typical (e.g., vesicourethral thickening), or atypical (e.g., liver), of PCa involvement, or in any lesion on a CT without tracer uptake that would require further workup.

### 2.3. Image Interpretation

^68^Ga-PSMA PET/CT scan images were reviewed by a nuclear medicine specialist and a radiology specialist. The two specialists were blinded to all other clinical and otherwise available imaging data, except postoperative PSA levels. If there were equivocal findings or discrepancies between the two readers, the final reading was decided in a consensus meeting. Individual lesions were categorized with PSMA-RADS version 2.0 (Appendix A). An overall score was obtained based on the highest individual lesion score. All lesions suggestive for recurrent PCa were noted and grouped into local recurrence, lymph-node metastases (divided into pelvic, retroperitoneal, and supradiaphragmatic location), bone metastases, and other metastases (e.g., lung, liver). For the analysis of detection rates, categories PSMA-RADS 3 to 5 were considered as “positive for malignancy” and 1–2 as “negative”. In patients with multiple lesions, the lesion with the highest score determined the overall score for the patient, except in the case of intense uptake in a site highly atypical for PCa (PSMA-RADS 3C).

### 2.4. Statistical Analysis

Data were entered into the Excel sheet (Microsoft Excel version 2405) manually and statistical analyses were performed using Excel.

## 3. Results

We included 128 consecutive PCa patients (mean age 70.4, range 55–87 years) with evidence of BR and a PSA level variable from 0.03 to 19 ng/mL (mean 1.3 ng/mL). Clinical characteristics of the 128 patients are presented in Table 1.

According to PSMA-RADS version 2.0, we classified the 128 patients as follows: 23 patients without PSMA-RADS score because they were completely negative. PSMA-RADS 1: 10 patients, PSMA-RADS 2: 4 patients, PSMA-RADS 3A: 11patients, PSMA-RADS 3B: 2 patients, PSMA-RADS 3C: 2 patients, PSMA-RADS 3D: 2 patients, PSMA-RADS 4: 13 patients, and PSMA-RADS 5: 61 patients.

***PSMA-RADS score 1.*** These 10 patients’ findings included biopsied thyroid nodules in 1 patient, hepatic hemangiomas in 2 patients, and adrenal adenomas in 2 patients confirmed by the features of contrast media multiphase CT. Cervicothoracic, coeliac, and sacral ganglia that may mimic lymph nodes were observed in three patients and lung peripheral fibrous changes and ground glass opacities (GGOs) in two patients, according to a recent history of COVID-19 infection.

***PSMA-RADS score 2.*** These four patients’ findings included axillary or hilar lymph nodes in two patients and bone degenerative changes in two patients.

***PSMA-RADS score 3A.*** These 11 patients’ findings included equivocal/low uptake at the vesicourethral anastomosis (VUA) in 3 patients (standardized uptake value max (SUVmax) range 1.8–2.5) (Figure 1), retrovesical region (RVR) in 1 patient (SUVmax 2.5), right iliac fossa in 1 patient (SUVmax 1.9), pelvic lymph nodes in 5 patients (SUVmax range 1–2.8), and mediastinum lymph node in 1 patient (SUVmax 3). The findings were confirmed with follow-up ^68^Ga-PSMA PET/CT in seven patients, with pelvic MRI in one patient, and with PSA level reduction after RT and/or ADT in three patients (Figure 1).

***PSMA-RADS score 3B.*** These two patients’ findings included low focal uptake without a corresponding lesion in CT in a rib in one patient (Figure 2) and in a vertebra in one patient. The findings were confirmed with follow-up ^68^Ga-PSMA PET/CT in both patients (Figure 2).

Biopsy revealed papillary thyroid carcinoma in the first patient and clear cell variant hepatocellular carcinoma. Both patients also performed follow-up ^68^Ga-PSMA PET/CT, and, in one of them, high tracer uptake in a perirectal adenopathy occurred (PSMA-RADS 4).

***PSMA-RADS score 3C.*** In these two patients, ^68^Ga-PSMA PET/CT showed intense uptake in the right lobe of the thyroid (Figure 3) and in the left lobe of the liver (Figure 4).

***PSMA-RADS score 3D.*** In these two patients, ^68^Ga-PSMA PET/CT detected multiple lung nodules in the former and a single lung nodule in the latter, both of which were considered to be metastases. In the first patient follow-up,^68^Ga-PSMA PET/CT showed an increased number of pulmonary nodules and, in the second, the appearance of radiotracer uptake and an increase in size. In both patients, the PSA was increased. No histopathology confirmation was available. The findings were confirmed with follow-up ^68^Ga-PSMA PET/CT in both patients

***PSMA-RADS score 4.*** These 13 patients’ findings included intense uptake at the VUA in 5 patients (SUVmax range 3.5–13) (Figure 5), retrovesical region in 3 patients (SUVmax range 3.5–4), pelvic lymph nodes in 4 patients (SUVmax range 3.8–8), and a rib in 1 patient (SUVmax 10) (Figure 5 and Figure 6).

Findings were confirmed with follow-up ^68^Ga-PSMA PET/CT in three patients, with pelvic MRI in one patient, and with PSA level reduction after RT and/or ADT in nine patients. In this group, four patients were positive in more than one region of interest (three with local recurrence, and pelvic lymph nodes, and one with local recurrence, pelvic lymph nodes, and bone metastases).

***PSMA-RADS score 5.*** These 61 patients’ findings included intense uptake at the VUA in 18 patients (SUVmax range 5–46), retrovesical region in 7 patients (SUVmax range 6–35), perirectal fat in 1 patient (SUVmax 40), presacral site in 1 patient (SUVmax 27), seminal vesicle bed in 2 patients (SUVmax range 8–9), pelvic lymph nodes in 13 patients (SUVmax range 4–85), retroperitoneal lymph nodes in 4 patients (SUVmax range 4.6–30), mediastinum and/or supraclavicular lymph nodes in 3 patients (SUVmax range 14–60), bone lesions in 10 patients (SUVmax range 6–51), and liver lesions in 2 patients (SUVmax 17–18). The findings were confirmed with follow-up ^68^Ga-PSMA PET/CT in 28 patients, with pelvic MRI in 9 patients, with whole-body CT follow-up in 1 patient (Figure 7), and with PSA level reduction after RT and/or ADT in 24 patients).

In this group, 11 patients were positive in more than one region of interest (5 with local recurrence and pelvic lymph nodes, 3 with pelvic lymph nodes and bone metastases and 3 with mediastinum and/or supraclavicular lymph nodes and bone metastases).

The overall detection rate of ^68^Ga-PSMA PET/CT was 71% (91/128).

Of the 37 cases classified as negative (PSMA-RADS 0–2), 32 underwent a second follow-up PSMA PET/CT because the PSA had further increased. In 11/32, lesions appeared that were not detectable even after a review of the first PET scan, and therefore these cases were considered as true negatives, even with imaging. The 13 cases classified as PSMA-RADS 3 (equivocal) and considered as positive in our study were also confirmed as true positives with a second follow-up ^68^Ga-PSMA PET/CT scan, as well as congruence with clinical data after salvage therapy. Moreover, the results of 31/74 cases classified as PSMA-RADS 4 or 5 were also confirmed by a second ^68^Ga-PSMA PET/CT scan, as well as congruence with clinical data after salvage therapy.

By dividing the patients into four groups according to PSA level before examination (PSA trigger), we obtained the following detection rates: PSA < 0.2 ng/mL (38%; 5/13), 0.2 ≤ PSA < 0.5 (57%; 20/35), 0.5 ≤ PSA ≤ 1 (77%; 31/40), and PSA > 1 (95%; 38/40).

The different regions with recurrent disease are listed in Table 2.

A total of 15 patients had two or three regions positive in the same examination. A total of 8 out of 20 patients had multiple bone metastases. Figure 8 correlates the sites of increased uptake of ^68^Ga-PSMA PET/CT with the corresponding PSMA-RADS category.

### Follow-Up

Follow-up was performed by monitoring PSA in 100/128 (78%) patients and with a second follow-up ^68^Ga-PSMA PET/CT in 76/128 patients (59%). The mean interval between the two PET/CT examinations was 11 months (range 6–49 months). The time course of the PSA values of patients rated as positive after salvage treatment or negative is outlined in Figure 9.

In the positive group of patients, 64% (48/76) showed a sufficient PSA response, whereas, in 36% (28/76), there was no sufficient PSA response. Of the 48 patients in whom PSA lowered after salvage treatment, 20 had local recurrence and 28 had adenopathy and/or metastasis. Of the 28 patients in whom PSA increased after salvage therapy, 1 had local recurrence and 27 had adenopathy and/or metastasis (Figure 9A). In the negative group of patients, the PSA level increased in 11 patients and decreased in 21 patients (Figure 9B).

## 4. Discussion

PSMA-RADS 1.0 was introduced to help recognize possible pitfalls and serve as a reference for scan interpretation [8]. PSMA-RADS 1.0 has been validated in several studies [9,10], but PSMA-RADS version 2.0 was recently introduced [7]. As far as we know, this is the first article that has used PSMA-RADS version 2.0. Briefly, the differences between the two versions are the subcategories in score 1 and mainly the subcategories in score 3. The 1A lesions were removed (benign by definition) and 1B lesions were directly classified as PSMA-RADS 1, being either notoriously benign based on their pathognomonic appearance in CT imaging or previously biopsied. PSMA-RADS 3 was significantly modified but the concept that these lesions need further investigation or follow-up remains. In the 3A category, equivocal uptake was seen in a soft-tissue site typical of PCa involvement, for example, pelvic or retroperitoneal lymph nodes. In the 3B category, the equivocal uptake was found in a bone lesion not definitive but also typical of PCa on anatomic imaging and can include a pure marrow-based lesion with little, if any, surrounding bony reaction, lytic or infiltrative lesion, or classic osteoblastic lesion. It is also suggested that follow-up imaging (anatomic or with PSMA PET/CT) showing progression can be decisive for diagnosis when a biopsy cannot be performed. In category 3A or 3B, it is up to the reader to decide whether these lesions should be reassigned and upgraded to PSMA-RADS 4 in oligometastatic patients (less than five lesions). PSMA-RADS 3D has been modified from lesions that are concerning for the presence of PCa or a non-prostate malignancy but lack radiotracer uptake to abnormal and suspicious lesions on a CT scan with no uptake of the PSMA ligand or otherwise lower than background. Consequently, this score is no longer restricted to lesions that are exclusively suspected for malignancy, as missing radiotracer uptake may be due to other reasons. Moreover, a new subcategory in PSMA-RADS 5 was introduced. The 5T subcategory includes treated lesions that were previously identified as metastases with or without uptake [7].

The authors found that the modifications in PSMA-RADS version 2.0 significantly improved scan interpretation, as many questionable findings, especially in PSMA-RADS 3, were clarified. Unfortunately, a significant number of patients with PCa cannot undergo biopsy for histopathological confirmation, so the verification of a certain finding with a follow-up PET scan is decisive. Last but not least, a standardized approach to PSMA PET/CT with PSMA-RADS version2.0 may be useful in anticipation of 177Lu-PSMA-617 treatment, but this aspect will need further verification.

The aim of this study was to assess the performance of ^68^Ga-PSMA-11 PET/CT, using PSMA-RADS 2.0. The overall detection rate of ^68^Ga-PSMA PET/CT in this study was 71%. Our results were sufficiently congruent with the recent literature regarding detection rates in relation to PSA values [11,12,13]. In the group of patients with PSA < 0.2 ng/mL, we obtained a positivity rate of 38%, 57% in the group of patients with 0.2 ≤ PSA < 0.5, 77% in the PSA 0.5 ≤ group, and 95% in the group of patients with PSA > 1. In a recent systematic review and meta-analysis [12] analyzing 64 studies with a total of 11,173 patients with BR, the PSMA PET/CT detection rates were 37% for PSA levels less than 0.5 ng/mL and 44% for PSA levels of 0.5–0.99 ng/mL. For a PSA level of 1–1.99 ng/mL, the detection rate was 61%. In another large multicentric study [13], analyzing 2533 patients with BR after RP, the detection rate of ^68^Ga-PSMA PET/CT was 43% for PSA ≤ 0.2 ng/mL, while for PSA > 0.2 to ≤0.5 ng/mL, the detection rate was 58%. Furthermore, for PSA > 0.5 to ≤1.0 ng/mL, the detection rate was 72% and increased to a maximum of 93% for PSA > 10 ng/mL. A recent systematic review and meta-analysis published in 2022 analyzed a total of 15 studies. Nine studies assessed the DR of 18F-PSMA-1007 PET/CT in BCR based on patient analysis without-serum PSA grouping, with a range of 47% to 95% and a pooled estimate of 82% (95% CI: 74% to 88%). Four studies [14,15,16,17] assessed the DR of 18F-PSMA-1007 PET/CT in a lesion-based without-serum PSA grouping, with a range of 33% to 100% and a combined estimate of 78% (95% CI: 33–100%) [18]. An interesting retrospective study analyzed 115 patients who underwent radical prostatectomy and showed BCR with increasing but still very low PSA levels (<0.2 ng/mL). This paper represents a 7-year “real-world” experience. The examiners were able to identify apparent oligometastatic disease with ^68^Ga-PSMA-11 PET/CT in nine patients (7.8%) at a PSA as low as 0.03 ng/mL. The scan positivity rates increased with PSA values > 0.15 ng/mL, a PSA doubling time of less than 12 months, or Gleason score 7b. The overall DRs were 21.1%, 18.9%, and 35% for PSA levels < 0.1 ng/mL, between 0.1 and 0.15 ng/mL, and >0.15 ng/mL, respectively [19].

The main limitation of our study is that due to practical and ethical issues, histopathological confirmation of the findings is missing. This is a common limitation of imaging studies in recurrent PCa patients. Still, our follow-up was very robust because in addition to the time course of PSA values, more than half of our patients underwenta second ^68^Ga-PSMA PET/CT (76/128).

## 5. Conclusions

In conclusion, using PSMA-RADS version 2.0, we obtained detection rate values comparable with recent literature both in absolute terms and in relation to different PSA levels. Further studies will be needed to validate PSMA-RADS version 2.0, including the validation of the PSMA-RADS 5T score (treated PCa metastasis), which was not explored in our study.

## Figures and Tables

**Figure 1 diagnostics-14-01291-f001:**
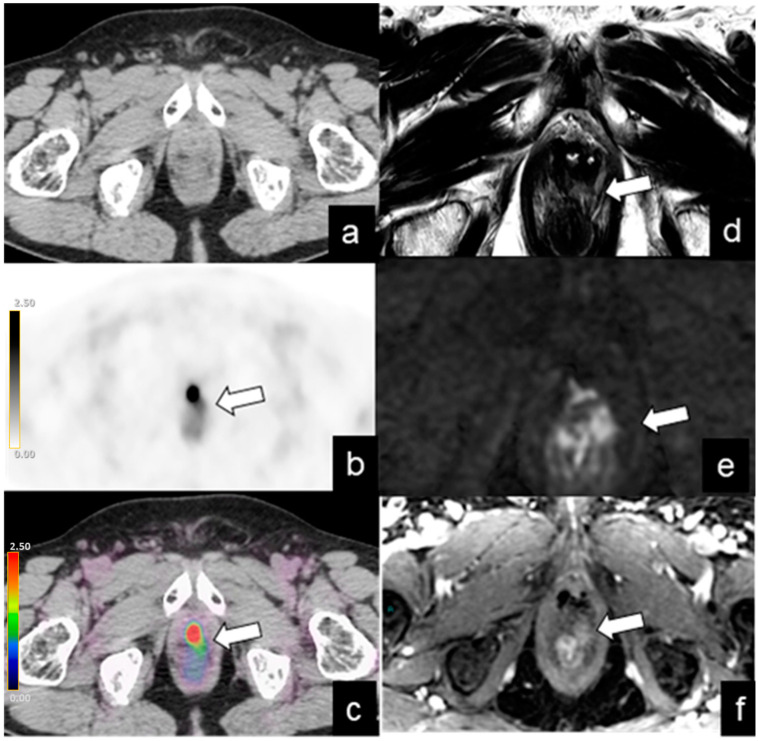
BR (PSA level = 2 ng/mL) in a 70-year-old patient 7 years after radical prostatectomy for pT3a N0, GS 8 (3 + 5) PCa. Axial PET (**b**) and fused PET/CT (**c**) images show focal and equivocal uptake (SUVmax 2.4) on the left side of the VUA (arrow), without morphological evidence in CT image (**a**), classified as PSMA-RADS 3A. T2-weighted (**d**), diffusion-weighted imaging (**e**), and dynamic contrast enhancement (**f**) MRI 3T images confirm diagnosis of local recurrence (arrow).

**Figure 2 diagnostics-14-01291-f002:**
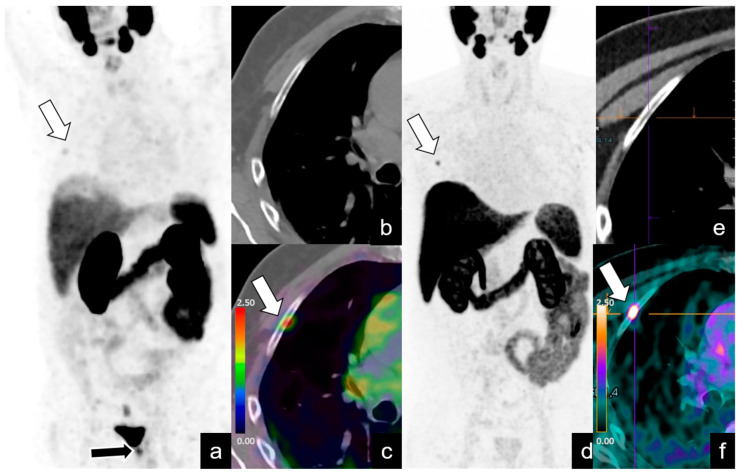
BR (PSA level = 2 ng/mL) in a 74-year-old patient 3 years after RP for pT2 N0, Gleason Score (GS) 7 (3 + 4) PCa. Maximum intensity projection (MIP) images (**a**) from ^68^Ga-PSMA PET, axial CT (**b**) and axial fused PET/CT (**c**) images at the level of the chest show focal uptake of the VUA (black arrow), consistent with local recurrence, and focal and equivocal uptake (SUVmax 2) of the right third rib (white arrow), classified as PSMA-RADS 3B, and initially considered by clinicians as benign. The patient underwent radiotherapy at the vesicourethral junction with the PSA level dropping to 0.02 ng/mL. Follow-up ^18^F-PSMA PET/CT (**d**), performed with a PSA level of 0.7 ng/mL, demonstrates the disappearance of radio treated recurrence at the vesicourethral junction and high uptake at the right third rib (SUVmax 8) (**e**,**f**). The lesion at the right third rib was treated with RT, and the PSA level decreased to 0.03 ng/mL.

**Figure 3 diagnostics-14-01291-f003:**
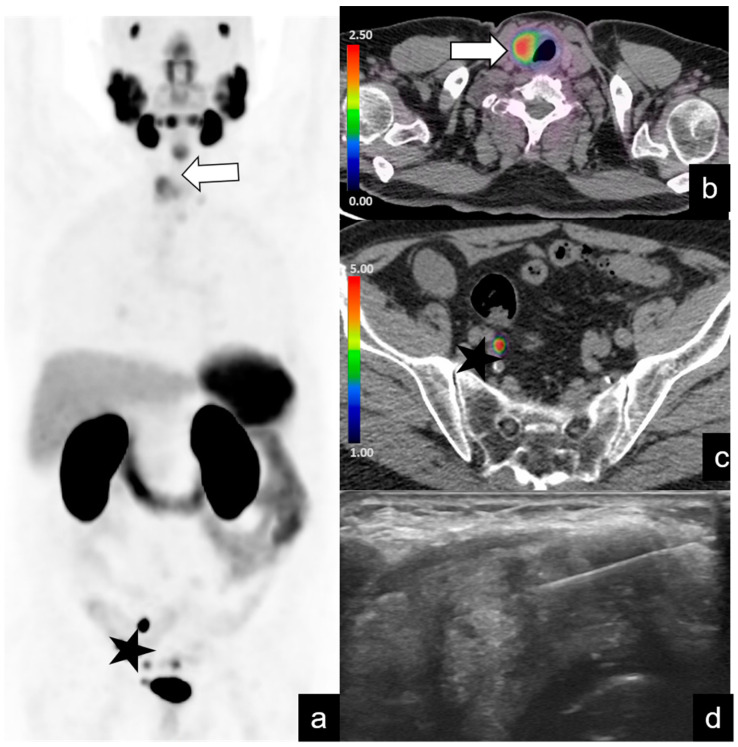
A72-year-old patient, already treated with RP and pelvic radiotherapy for PCa (pTb N1, GS 9 [4 + 5]), performed PET/CT when the PSA level increased during hormone therapy. PET/CT images (**a**–**c**) demonstrated the presence of right external iliac adenopathy (asterisk in (**a**,**c**)) and intense right lobe thyroid uptake (arrow in (**a**,**b**)), suspicious for thyroid cancer and classified as PSMA-RADS 3C. Echo-guided biopsy (**d**) confirmed the diagnosis of papillary thyroid carcinoma.

**Figure 4 diagnostics-14-01291-f004:**
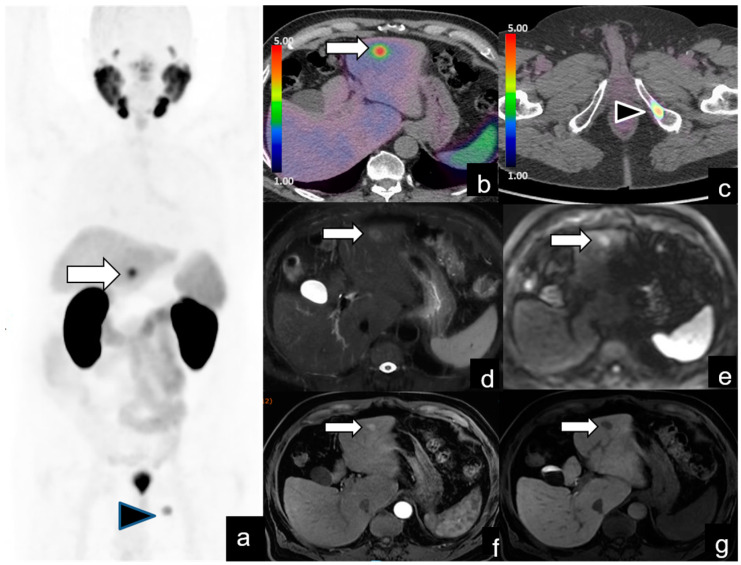
BR (PSA level = 1.4 ng/mL) in a 70-year-old patient 2 years after RP for pT2c N0, GS 8 (4 + 4) prostate cancer. PET/CT images demonstrate the presence of a bone lesion at the left iliac bone (arrowhead in (**a**,**c**)) and intense uptake in the left lobe of the liver (arrow in (**a**,**b**)), suspicious for primary hepatic tumor and classified as PSMA-RADS 3C. MRI images (Fat Sat T2-weighted (**d**), diffusion-weighted imaging (**e**), arterial (**f**) and hepatobiliary (**g**) contrast enhancement phases) are consistent with a primary hepatic tumor, confirmed by biopsy and treated with left hepatectomy. Definitive histological diagnosis was clear cell variant hepatocellular carcinoma.

**Figure 5 diagnostics-14-01291-f005:**
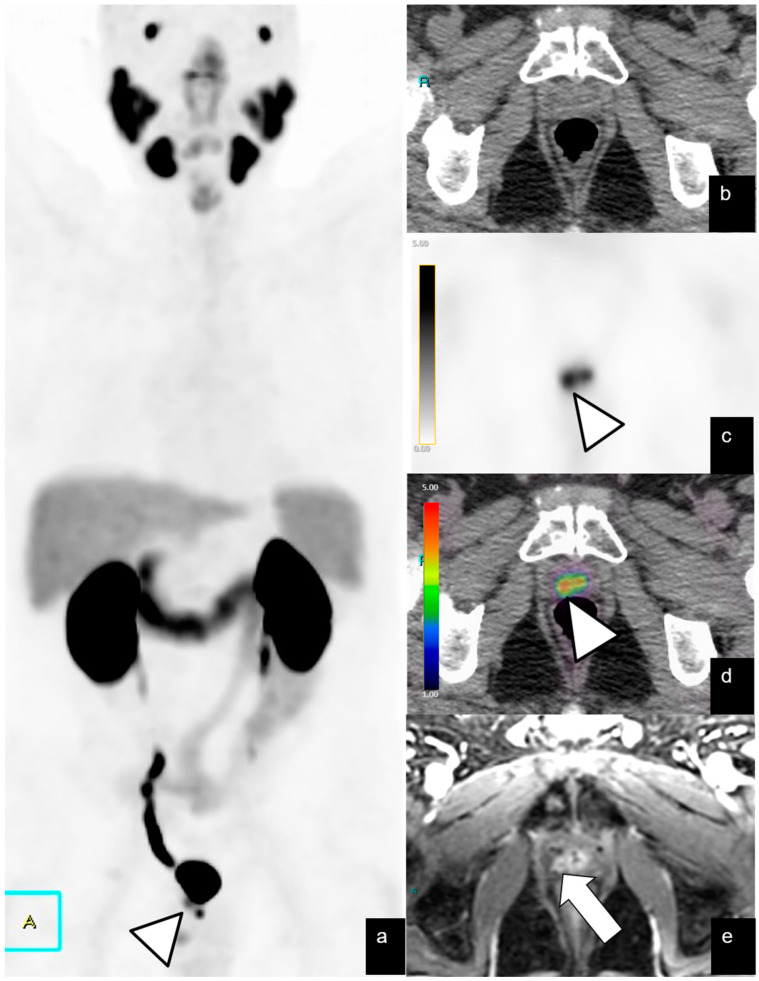
BR (PSA level = 1.1 ng/mL) in a 70-year-old patient 7 years after RP for pT2c N0, GS 7 (3 + 4) PCa. PET/CT MIP (**a**), axial PET (**c**), and fused PET/CT (**d**) images show focal and intense uptake (SUVmax 12) on the right side of the VUA (arrowhead), without morphological evidence in CT image (**b**), classified as PSMA-RADS 4. Axial T1-weighted FS contrast enhancement MRI 3T image (**e**) confirmed a diagnosis of local recurrence (arrow).

**Figure 6 diagnostics-14-01291-f006:**
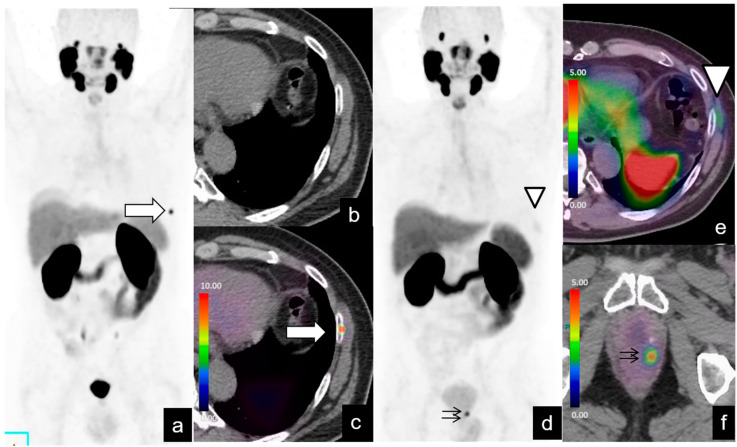
BR (PSA level = 0.19 ng/mL) in a 65-year-old patient, 1 year after RP for pT3 N0, GS 8 (4 + 4) prostate cancer. PET/CT images (**a**,**c**) demonstrate intense focal uptake (SUVmax 13) at the left seventh rib (arrow) without changes on CT (**b**), classified as PSMA-RADS 4. The patient underwent radiotherapy of the left seventh rib, and the PSA level dropped to 0.07 ng/mL. Follow-up PSMA PET/CT, performed with a PSA level of 0.18 ng/mL, showed a disappearance of uptake at the left seventh rib with evidence of adjacent post-RT myositis (arrowhead in (**d**,**e**)) and focal uptake (SUVmax 5.2) in the left paramedian retrovesical site (**d**,**f**), related to PSMA-RADS 4.

**Figure 7 diagnostics-14-01291-f007:**
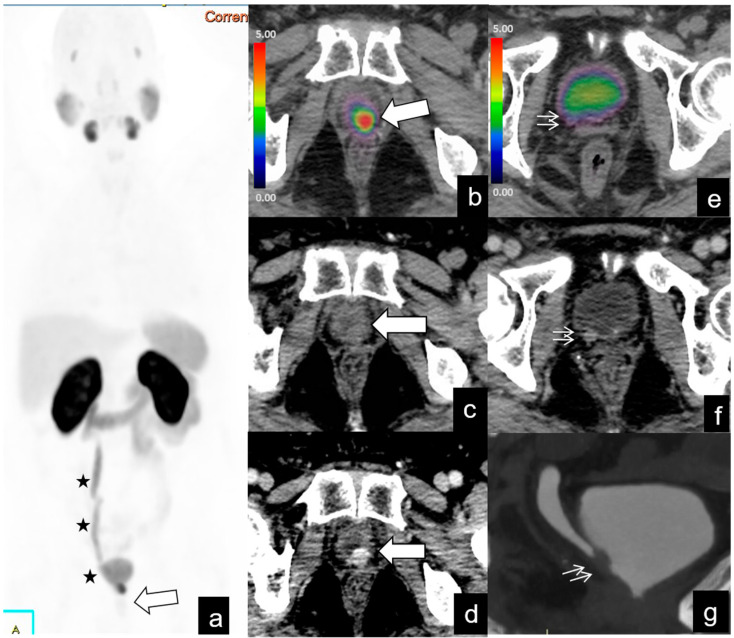
BR (PSA level = 0.2 ng/mL) in a 64-year-old patient, 7 years after RP for pT3 N0, GS 9 (4 + 5) PCa, and 1 year before trans-urethral resection (TURB) of right posterior wall of the bladder with definitive histological diagnosis of metastasis from PCa. MIP PET/CT (**a**), axial fused PET/CT (**b**), axial CT (**c**), and axial contrast media CT (**d**) showed a nodule with intense tracer uptake (SUVmax 41) and strong vascularization (arrow) at the left paramedian retrovesical midline (PSMA-RADS 5), infiltrating the posterior bladder wall, without hydronephrosis. PET/CT images (**a**,**e**) and contrast media CT images during the venous (**f**) and excretory (**g**) phases show moderately vascularized thickening (arrows in (**e**)), without tracer uptake (arrows in (**d**)) at the level of the right posterior wall, consistent with post-surgical fibrotic changes (previous TURB), responsible for hydroureteronephrosis (asterisks).

**Figure 8 diagnostics-14-01291-f008:**
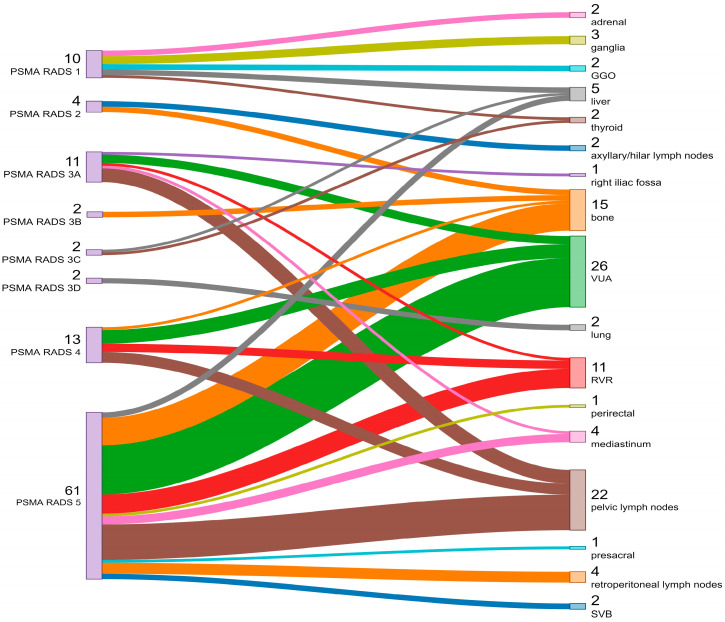
Correlation between sites of increased uptake and PSMA-RADS categories. GGO—ground glass opacity; VUA—vesicourethral anastomosis; RVR—retrovesical region; SVB—seminal vesicles bed.

**Figure 9 diagnostics-14-01291-f009:**
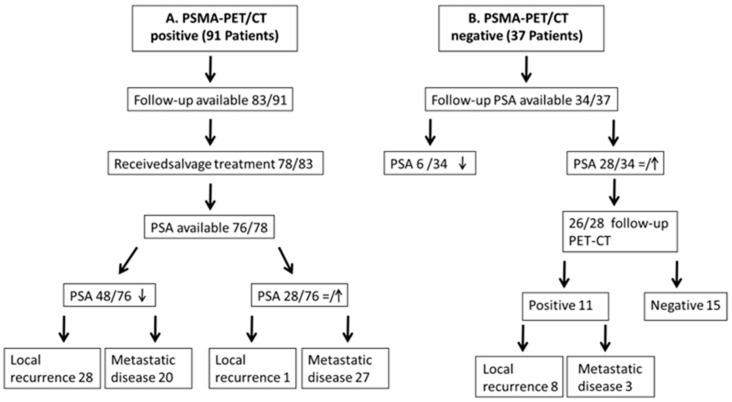
Follow-up of patients evaluated as PSMA-PET/CT positive (**A**) and negative (**B**). Arrows within the boxes indicate PSA trend.

**Table 1 diagnostics-14-01291-t001:** Patients’ Characteristics.

	All Patients, N = 128	Positive Scans, N = 91	Negative Scans, N = 37
**Age**			
Median	70.4	70.8	68.5
Range	55–87 years	55–87 years	55–85 years
**Primary Gleason score**			
Median	7	7	7
Range	5–9	5–9	6–9
**T stage**			
	5 pT2a, 14 pT2b, 36 pT2c, 42 pT3a, 31pT3b	3 pT2a, 10 pT2b, 15 pT2c, 22 pT3a, 21 pT3b	2 pT2a, 4 pT2b, 21 pT2c, 20 pT3a, 10 pT3b
Range	pT2a–pT3b	pT2a–pT3b	pT2a–pT3b
**PSA (ng/mL)**			
Median	1.3 ng/mL	1.6 ng/mL	0.5 ng/mL
Range	0.03–19 ng/mL	0.05–19 ng/mL	0.03–1.4 ng/mL
**Further treatment**			
External radiation after RP	40	24	16
Lymphadenectomy	5	4	1
Anti-hormonal treatment	13	9	4

**Table 2 diagnostics-14-01291-t002:** Sites with PCa recurrence in ^68^Ga-PSMA PET/CT.

Region	No. of Patients (%)	SUVmax Median	SUVmax Range	Size (mm) Median	Size (mm) Range
**Local recurrence**	42/128 (32%)	12	2.1–46	9.6	3–53
**Lymph-node metastases**					
Abdominopelvic	39/128 (30%)	24	1–73	7.3	2–19
Supradiaphragmatic	4/128 (3%)	27.2	3–60	18.5	15–22
**Bone metastases**	20/128 (15%)	19	1.9–70		
**Other (e.g., lung, liver) metastases**	5/128 (3%)	14.6	2.5–17		

## Data Availability

The data presented in this study are available on request from the corresponding author due to patients’ consent.

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
