# Peer review of "68Ga-PSMA PET/CT in Recurrent Prostate Cancer after Radical Prostatectomy Using PSMA-RADS Version 2.0"

_diagnostics, 2024, doi:10.3390/diagnostics14121291_

Round 1

Reviewer 1 Report

Comments and Suggestions for Authors

Thank you for the opportunity to review the manuscript evaluated 68Ga-PSMA PET/CT In Recurrence Prostate Cancer After Radical Prostatectomy Using PSMA-RADS version 2.0. The work is interesting and provides clinically interesting information, but needs some improvements. 

For the name of the radiopharmaceutical, could the authors use nomenclature consistent with EANM guidelines?

Line 46 - For a PSA range of 0.5-1.0  according to the literature, is the sensitivity of PET/CT exactly 72.2%?

Table S1 should be included to the main text, as provide basic for this manuscript information.

Line 161 Could the authors add information if for patients with PSMA – RADS score 3D was the histopathology verification available?

Significant part of  the discussion provides results which should be shifted (after modification) to the proper section. It would be clearer if the sensitivity of the PET/CT scan in accordance to PSA levels were summarized in a table.

The discussion section should be rewritten, as currently it mainly contains information about the results, rather than a discussion of them.

Author Response

- We corrected the name of the radiopharmaceutical. 

- Line 46 -  Thank You for your comment, we revised the values.

- We included Table S1 as supplementary material following our editor suggestion.

- Line 161 No histopathology confirmation was available, we specified this concept in the results section. the results were confirmed with follow up PSMA PET scan, we also specified this aspect more clearly.

- Following your suggestion we moved the results in the correct section and revised the discussion.

Reviewer 2 Report

Comments and Suggestions for Authors

I would like to thank the authors for this interesting manuscript.

I think the ambition is to high if you want to assess the performance of a scoring system and then subdived in so many categories. I think it would be most interesting to see how the performance is if the patients are group in negative and positive. A comparison with PI-RADS version 1.0 could also help to show if version 2.0 is performing better.

A graphical depiction of classification (sankey diagram? maybe with the subdivion in PI-RADS) with the final outcome prostate cancer vs others light help to visualise the performance. 

The table (2) is difficult to read and should be restructured. Table 1 is missing?

The purpuse of figure A is not very clear to me. If the PI-RADS could be incorporated it would mean more I would say. Other figure are numbered, why this difference? 

Comments on the Quality of English Language

only very minor spelling correction needed

Author Response

- Comparing the performance between PSMA-RADS version 1 and version 2 is beyond the scope of this study. A small comparison between the 2 versions has been included in the discussion section.

- Former figure A (now diagram B)  splits patients in positive and negative at PSMA PET scan.  Following your suggestion we included a Sankey diagram to help clarify different PSMA RADS 2.0 categories and pathological finding.

- For table (2) we used a template similar to other papers found in literature. We kindly ask You if you have any specifical suggestion to make the table more understandable.

.- We revised the name of the Tables.

- Following your suggestion we changed the name from Figure A to Diagram B. We also included a Sankey diagram in order to make the results clearer.

Round 2

Reviewer 1 Report

Comments and Suggestions for Authors

Thank you, for the corrections, the authors have made all the necessary amendments. 

Author Response

Thank You for your revision. 

Reviewer 2 Report

Comments and Suggestions for Authors

I would like to thank the authors for their responses.

One small detail, a colour bar with a SUV scale next to the PET images would help the interpretation. I assume they are all scaled the same, but nevertheless...

Author Response

Thank You for your revision, we added a colour bar with SUV scale next to the PET images.